# SimULi: Real-Time LiDAR and Camera Simulation with Unscented Transforms

**Haithem Turki**[*]  **Qi Wu**[*]  **Xin Kang**  **Janick Martinez Esturo**

**Shengyu Huang**  **Ruilong Li**  **Zan Gojcic**  **Riccardo de Lutio**

NVIDIA
{hturki,qiwu,rdelutio}@nvidia.com
https://research.nvidia.com/labs/sil/projects/simuli

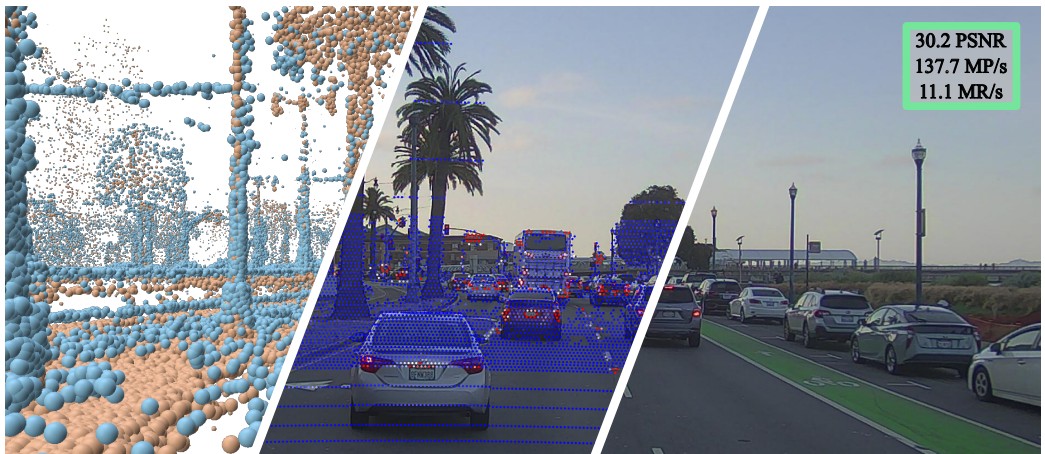

Figure 1: **SimULi.** We design a factorized 3D Gaussian representation that encodes camera and LiDAR information into separate sets of 3D Gaussians joined via nearest-neighbor anchoring loss (**left**). To efficiently render LiDAR scans (**middle**), we extend 3DGUT (Wu et al., 2025b) with an automated tiling strategy and ray-based culling. When compared to existing methods, we render 1.5-20× faster and match or exceed LiDAR and camera quality (**right**) across a wide range of metrics.

## ABSTRACT

Rigorous testing of autonomous robots, such as self-driving vehicles, is essential to ensure their safety in real-world deployments. This requires building high-fidelity simulators to test scenarios beyond those that can be safely or exhaustively collected in the real-world. Existing neural rendering methods based on NeRF and 3DGS hold promise but suffer from low rendering speeds or can only render pinhole camera models, hindering their suitability to applications that commonly require high-distortion lenses and LiDAR data. Multi-sensor simulation poses additional challenges as existing methods handle cross-sensor inconsistencies by favoring the quality of one modality at the expense of others. To overcome these limitations, we propose SimULi, the first method capable of rendering arbitrary camera models and LiDAR data in real-time. Our method extends 3DGUT, which natively supports complex camera models, with LiDAR support, via an automated tiling strategy for arbitrary spinning LiDAR models and ray-based culling. To address cross-sensor inconsistencies, we design a factorized 3D Gaussian representation and anchoring strategy that reduces mean camera and depth error by up to 40% compared to existing methods. SimULi renders 10-20× faster than ray tracing approaches and 1.5-10× faster than prior rasterization-based work (and handles a wider range of camera models). When evaluated on two widely benchmarked autonomous driving datasets, SimULi matches or exceeds the fidelity of existing state-of-the-art methods across numerous camera and LiDAR metrics.

---

[*]Equal contribution

# 1 INTRODUCTION

With the rise of end-to-end policy models, accurate sensor simulation has become a critical component in the development and evaluation of autonomous vehicle (AV) systems. Data-driven methods based on Neural Radiance Fields (NeRFs) (Mildenhall et al., 2020) offer a scalable solution for reconstructing high-quality, diverse simulation environments directly from real-world sensor data (Turki et al., 2023; Yang et al., 2023b). As they are optimized to match real-world observations, they also exhibit a smaller domain gap compared to traditional artist-generated simulators.

**Multi-Sensor Simulation.** In addition to cameras, most AVs are equipped with active sensors such as LiDAR to obtain 3D measurements of their surroundings. Early approaches focus on camera simulation (Rematas et al., 2022), using other sensors to constrain geometry for improved novel view synthesis, or solely on LiDAR (Tao et al., 2023; Huang et al., 2023). Recent efforts (Yang et al., 2023b; Tonderski et al., 2024; Hess et al., 2025) jointly model both sensors, a challenging task as cross-sensor calibration inconsistencies due are impossible to eliminate in the real world. In practice, they are forced to prioritize camera quality at the cost of LiDAR accuracy (or vice-versa).

**Sensor Rendering.** NeRF-based methods (Yang et al., 2023b; Tonderski et al., 2024) suffer from low rendering speeds, limiting their practicality for large-scale use. Replacing NeRFs with a 3D Gaussian Splatting (3DGS) formulation (Kerbl et al., 2023) mitigates these challenges by enabling real-time rendering through a rasterization-based pipeline, while maintaining, or even improving, visual fidelity, especially for larger scenes (Chen et al., 2025; Yan et al., 2024; Zhou et al., 2024).

However, this comes at the cost of limitations inherent to the rasterization paradigm. In particular, rasterization is constrained to idealized pinhole camera models and does not naturally support non-linear projection functions (e.g., fisheye lenses) or time-dependent effects such as rolling shutter, both of which are common in data captured by AV sensor suites (Xiao et al., 2021; Sun et al., 2020; Caesar et al., 2020; Wilson et al., 2021; Liao et al., 2021). Rectifying training images and ignoring rolling shutter effects degrades reconstruction quality, while the inability to render non-pinhole camera views introduces a significant domain gap for downstream models, which are typically trained on raw sensor data. LiDAR sensors are often neglected, as their sparse, irregular sampling pattern and strong time-dependent effects make them difficult to model within the 3DGS framework.

**SimULi.** In this work, we propose a high-fidelity and efficient reconstruction pipeline that enables joint camera and LiDAR simulation for AV scenarios. Importantly we aim to: **(i)** natively support arbitrary camera and LiDAR models, including their time-dependent effects; **(ii)** accurately model sensor characteristics (such as color, LiDAR intensity, and ray drop effects), without prioritizing the accuracy of one sensor at the expense of others as in existing work; and **(iii)** achieve real-time inference while maintaining high visual fidelity.

We build upon the 3D Gaussian Unscented Transform (3DGUT) (Wu et al., 2025b), which inherently supports non-linear camera models and time-dependent effects such as rolling shutter. In this work, we extend it further to support rotating LiDAR sensors with irregular sampling patterns. Specifically, we **(i)** derive a LiDAR measurement model that accounts for its time-dependent behavior, **(ii)** introduce a histogram-equalization-based algorithm to compute an optimal tiling scheme for handling the highly irregular structure of LiDAR measurements, and **(iii)** take advantage of the relative sparsity of LiDAR rays via a culling strategy that further accelerates rendering.

To accurately model multi-sensor data, we encode the information relevant to each modality into distinct sets of 3D Gaussians coupled via an efficient nearest-neighbor anchoring method. We derive several benefits. First, we find that this handles cross-sensor inconsistencies far more robustly than prior methods that encode all sensor data into a single NeRF (Yang et al., 2023b; Tonderski et al., 2024) or set of Gaussian particles (Hess et al., 2025), allowing us to match or exceed the accuracy of camera-only or LiDAR-only simulation. Second, this improves rendering efficiency, as the information needed to render each sensor is contained within a subset of Gaussians (accelerating LiDAR rendering by 2-3×). Third, by specializing the characteristics of each Gaussian set, we benefit from inductive biases inherent to each sensor. Since LiDAR returns typically intersect surfaces, we encourage sparsity and binary opacity in LiDAR Gaussians, further improving rendering speed. Camera Gaussians remain unconstrained and benefit from the smooth optimization properties and high rendering quality of volumetric approaches (Wang et al., 2023; Turki et al., 2024).

**Contributions.** We make the following contributions: (1) we extend 3DGUT with LiDAR support and introduce an automated tiling scheme from which we derive optimal tiling parameters for any spinning LiDAR model without relying on handcrafted heuristics (Hess et al., 2025), (2) we design a factorized 3D Gaussian representation and anchoring strategy that mitigates the camera vs LiDAR accuracy tradeoff needed in prior methods, (3) we present results that surpass existing LiDAR and camera-based SOTA, without relying on neural networks for refinement as in prior NeRF (Tonderski et al., 2024; Yang et al., 2023b; Zheng et al., 2024) and 3DGS-based work (Zhou et al., 2025; Hess et al., 2025). Our simplified approach renders 1.5-10× faster than recent rasterization-based methods (Hess et al., 2025), 10-20× faster than ray tracing approaches (Zhou et al., 2025), and is, to our knowledge, the first to render in real-time and support arbitrary camera and LiDAR models.

## 2 RELATED WORK

Recent neural reconstruction methods (Mildenhall et al., 2020; Kerbl et al., 2023) have greatly influenced AV simulation by enabling the reconstruction of large, high-fidelity environments from raw sensor data. We list a non-exhaustive selection of these methods along axes most relevant to ours.

**Simulating Camera Observations.** Early neural reconstruction methods in the AV domain focused on modeling camera observations (Ost et al., 2021; Xie et al., 2023; Guo et al., 2023; Yang et al., 2023a), often leveraging LiDAR data to constrain geometry and enhance novel view synthesis. To adapt static NeRF representations to the AV setting, these approaches introduced domain-specific extensions such as dynamic scene graph parameterizations (Ost et al., 2021) and specialized sky representations (Guo et al., 2023; Yang et al., 2023a). With the advent of 3DGS (Kerbl et al., 2023), which enables faster rendering and better scalability, several of these ideas have been reimplemented within particle-based representations (Chen et al., 2025; Yan et al., 2024; Chen et al., 2023). While 3DGS-based methods yield faster frame rates, they also inherit all the limitations of rasterization.

**Simulating LiDAR Observations.** In parallel to camera-focused approaches, several works have adapted neural reconstruction methods for LiDAR novel view synthesis (Huang et al., 2023; Tao et al., 2023; Zhang et al., 2024). NFL (Huang et al., 2023) introduced a physically grounded way of modeling LiDAR characteristics within NeRFs. DyNFL (Wu et al., 2024) and LiDAR4D (Zheng et al., 2024) extended NFL to dynamic scenes using the same scene graph parameterization as camera methods and improve rendering speed via acceleration structures (Müller et al., 2022).

Adapting 3DGS for LiDAR rendering is more challenging, as its rasterization assumes ideal pinhole camera models and its tiling strategy does not naturally accommodate the non-uniform sampling patterns of LiDAR sensors. Recently, 3DGRT (Moenne-Loccoz et al., 2024) replaced rasterization with a ray tracing formulation, enabling support for more complex sensor models. LiDAR-RT (Zhou et al., 2025) applies this framework specifically to LiDAR simulation, while GS-LiDAR (Jiang et al., 2025) combines a similar method with tile-based panoramic rendering. Both approaches render faster than NeRF-based approaches but remain several times slower than rasterization methods.

**Joint LiDAR-Camera Simulation.** Recent works jointly model LiDAR and camera novel view synthesis via a unified neural representation. UniSim (Yang et al., 2023b) and NeuRAD (Tonderski et al., 2024) build upon NeRF and adapt the rendering formulation for different sensors using the same underlying model. AlignMiF (Tang et al., 2024) encodes sensors into separate feature grids aligned via shared initialization and feature fusion. They remain impractical due to the slow rendering speed of NeRF and (other than AlignMiF) perform worse than camera-only or LiDAR reconstruction due to cross-sensor inconsistencies.

Closest to our work, SplatAD (Hess et al., 2025) extends 3DGS with a LiDAR rendering formulation that supports non-uniform sampling patterns (as does ours) and support for rolling shutter effects. While it addresses several important challenges, it differs from our method in three key aspects. First, as a 3DGS-based method, it remains limited to perfect pinhole camera models, whereas SimULi builds upon 3DGUT (Wu et al., 2025b), which enables native support for high-distortion camera models, such as fisheye lenses. Second, it relies on manual heuristics to determine the tiling strategy for each LiDAR sensor. In contrast, we demonstrate how an automated strategy can adaptively compute optimal tiling patterns across a wide range of non-uniform LiDAR sensors. Third, it

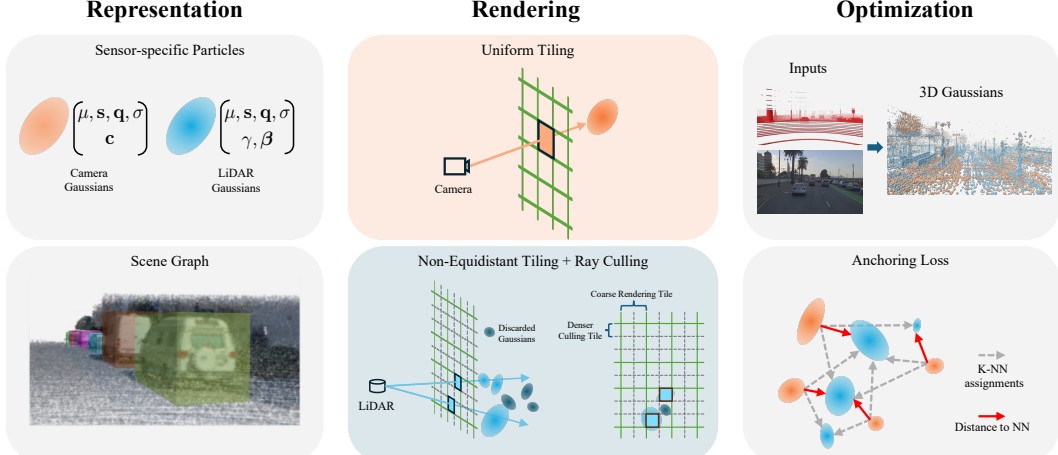

Figure 2: **Method Overview.** We model the scene as a dynamic graph (Ost et al., 2021) and parameterize the background and each actor with camera and LiDAR 3D Gaussians (**left**). We render camera views similar to 3DGUT (Wu et al., 2025b) and derive an automated tiling strategy and ray-based culling to efficiently render LiDAR (**middle**). We sample an image and LiDAR scan at each training step to optimize our representation (**right**). To improve camera novel view synthesis with LiDAR-supervised geometry, we anchor camera Gaussians near surfaces via nearest-neighbor loss.

encodes all sensor information into the same Gaussian set, and suffers from cross-sensor inconsistencies as prior NeRF-based works (Tonderski et al., 2024; Yang et al., 2023b), which we mitigate by encoding each sensor into its own particle set and using an anchoring loss. We illustrate the difference in approaches via extensive benchmarking in Sec. 4.

## 3 METHOD

Our goal is to learn a controllable scene representation that simulates camera and LiDAR renderings from novel viewpoints in real-time (Fig. 2). We describe our representation in Sec. 3.1, our general rendering approach in Sec. 3.2, LiDAR specifically in Sec. 3.3, and optimization in Sec. 3.4.

### 3.1 REPRESENTATION

**Particle Representation.** We encode camera and LiDAR attributes into separate unordered sets $G_c$ and $G_l$ of semi-transparent 3D Gaussian particles (Kerbl et al., 2023). Particles in both sets are parameterized by their 3D position $\boldsymbol{\mu}_i \in \mathbb{R}^3$, opacity coefficient $\sigma \in \mathbb{R}$, and covariance matrix $\boldsymbol{\Sigma}_i \in \mathbb{R}^{3\times3}$. $\boldsymbol{\Sigma}_i$ is decomposed into a rotation matrix $\mathbf{R} \in \mathrm{SO}(3)$ and a scaling matrix $\mathbf{S} \in \mathbb{R}^{3\times3}$, such that $\boldsymbol{\Sigma} = \mathbf{R}\mathbf{S}\mathbf{S}^T\mathbf{R}^T$, to ensure that it remains positive semi-definite. We associate $3^{rd}$-order spherical harmonics coefficients $\mathbf{SH^c} \in \mathbb{R}^{48}$ to camera Gaussians to encode view-dependent color, and $\mathbf{SH^l} \in \mathbb{R}^{48}$ to LiDAR Gaussians for view-dependent intensity and ray drop information.

**Scene Representation.** We adopt a graph decomposition similar to prior work (Ost et al., 2021) to enable controllability in dynamic scenes. We assign each particle in $G_c$ and $G_l$ to a dynamic object or the static background. Each object is associated with a 3D bounding box and a sequence of $\mathrm{SE}(3)$ poses adjusted with learnable offsets. The means and covariances of the Gaussians assigned to objects are expressed in the objects' local coordinate system. At render time, they are transformed to world coordinates by applying the $\mathrm{SE}(3)$ transformation corresponding to the timestamp $t$.

### 3.2 RENDERING

We build our differentiable renderer upon 3DGUT (Wu et al., 2025b), as it supports complex cameras and time-dependent effects that are common in autonomous driving data.

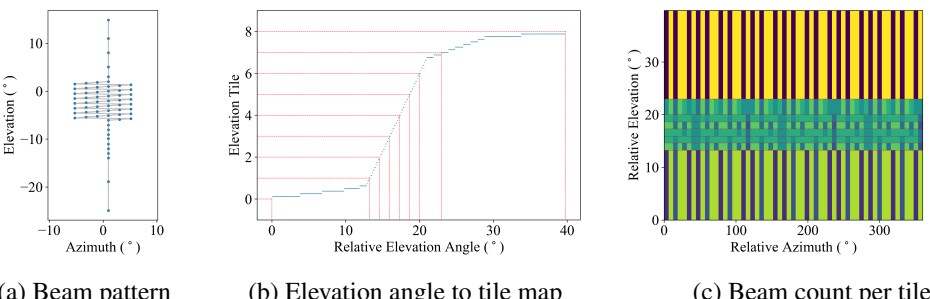

(a) Beam pattern        (b) Elevation angle to tile map        (c) Beam count per tile

Figure 3: **LiDAR Tiling.** As the measurement pattern of commonly used LiDAR sensors (Xiao et al., 2021) is irregular (**left**), rendering with equally spaced tiles is highly inefficient. We compute the normalized CDF of elevation angles using a predefined histogram bin count and set elevation tiling boundaries at angles where the CDF values cross integer boundaries (**middle**). We then compute an azimuth tile count such that the beam count per tile differs at most by 8 samples (**right**).

**Volume Rendering.** Given a camera ray $\mathbf{r}(\tau) = \mathbf{o} + \tau\mathbf{d}$ with origin $\mathbf{o} \in \mathbb{R}^3$ and direction $\mathbf{d} \in \mathbb{R}^3$, we volumetrically render a foreground color $\mathbf{c_f} \in \mathbb{R}^3$ and opacity $\omega \in \mathbb{R}$ via numerical integration over the interacting Gaussian particles in $G_c$:

$$\mathbf{c_f}(\mathbf{o}, \mathbf{d}) = \sum_{i \in G_c} \mathbf{SH}_i^{\mathbf{c}}(\mathbf{d})\alpha_i T_i, \quad \omega(\mathbf{o}, \mathbf{d}) = \sum_{i \in G_c} \alpha_i T_i, \quad \alpha_i = \sigma_i \rho_i(\mathbf{o} + \tau\mathbf{d}), \quad T_i = \prod_{j=1}^{i-1} 1 - \alpha_j,$$

(1)

where $\rho_i$ is the particle response described in the next paragraph. We alpha-composite the foreground color $\mathbf{c_f}$ with a background color $\mathbf{c_b}$ obtained from a learned texture environment map (Chen et al., 2023; 2025). We obtain the final color prediction $\mathbf{c}$ via an affine transformation from a learned bilateral grid $\mathcal{A}$ (Wang et al., 2024) that handles lighting variations across frames and cameras:

$$\mathbf{c}(\mathbf{o}, \mathbf{d}) = \mathcal{A}(\omega(\mathbf{o}, \mathbf{d})\mathbf{c_f}(\mathbf{o}, \mathbf{d}) + (1 - \omega(\mathbf{o}, \mathbf{d}))\mathbf{c_b}(\mathbf{d}))$$

(2)

We similarly render LiDAR features $\zeta \in \mathbb{R}^3$ from particles in $G_l$ as $\zeta(\mathbf{o}, \mathbf{d}) = \sum_{i \in G_l} \mathbf{SH}_i^{\mathbf{l}}(\mathbf{d})\alpha_i T_i$. We decode beam intensity $\gamma \in \mathbb{R}$ from the first channel of $\zeta$ and derive a ray drop probability $\beta \in \mathbb{R}$ by applying the softmax function on the remaining two channels ($\beta_{\text{hit}}, \beta_{\text{drop}}$) of $\zeta$ (Zhou et al., 2025).

**Particle Contributions and Response.** As in 3DGUT (Wu et al., 2025b), we determine which particles contribute to each ray by approximating each Gaussian via the Unscented Transform (UT). We project 7 sigma points per Gaussian to estimate a 2D conic $\Sigma'$ before applying tiling and culling as in 3DGS (Kerbl et al., 2023). As sigma points are independently projected, we trivially handle time-dependent effects such as rolling-shutter by incorporating camera motion into the projection function (Fig. 11). We measure the particle response function of each Gaussian in 3D via the distance $\tau_{max} := \text{argmax}_\tau \rho_i(\mathbf{o} + \tau\mathbf{d})$ that maximizes $\rho_i$ along the ray $\mathbf{r}(\tau)$ (Moenne-Loccoz et al., 2024).

### 3.3 LIDAR RENDERING

**Projection.** For a given LiDAR Gaussian in $G_l$, we project each of its 7 sigma points onto a 2D azimuth/elevation grid. We convert the 3D Euclidean coordinates of each sigma point from the world frame to the sensor frame and derive the corresponding spherical coordinates:

$$\phi = \text{arctan2}(y, x), \quad \omega = \arcsin(z/r), \quad r = \sqrt{x^2 + y^2 + z^2}$$

(3)

where $\phi$ denotes the azimuth, $\omega$ the elevation and $r$ the range. We derive a 2D conic from the projected coordinates to use for tiling and culling, and handle time-dependent effects (such as the LiDAR-equivalent of rolling shutter) in the same manner as with camera rendering.

**Tiling.** Effectively distributing work across tiles is crucial to achieving high rendering speeds. However, unlike cameras, the resolution of LiDAR measurements is irregular and differs greatly across dimensions (Fig. 3a). Although azimuths cover the full 360° radius, the elevation angle

**Depth Loss (λ=0.001)**
↑**PSNR: 29.94,** ↓**CD: 0.46**

**Depth Loss (λ=0.1)**
↑**PSNR: 27.53,** ↓**CD: 0.23**

**Anchoring Loss (Ours)**
↑**PSNR: 30.09,** ↓**CD: 0.22**

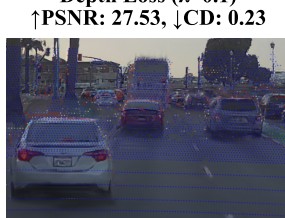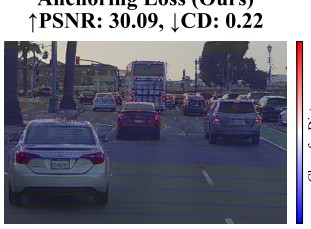

Figure 4: **Depth Supervision.** Prior work encodes camera and LiDAR into the same representation constrained with a LiDAR-supervised depth loss. As cross-sensor data is not fully consistent, this forces the representation to prioritize camera instead of LiDAR quality (**left**) or the inverse (**middle**), as shown by PSNR and chamfer distance. We factorize each modality into its own particle set joined via nearest-neighbor anchoring loss, improving the quality of each (**right**).

is usually much smaller (around $20°$). Using equally sized tiles as with cameras is thus highly inefficient. We derive an automated strategy that takes elevation angles as input and computes an equalized elevation tiling and azimuth tile count such that each elevation tile contains roughly the same number of measurements and that the count per tile is under a user-defined constraint $M$.

We compute the normalized cumulative distribution function $C$ using a pre-defined histogram bin count $r = 400$ (Fig. 3b). We then set elevation tiling boundaries at elevation angles where the corresponding cumulative distribution function values cross integer boundaries. We finally compute the azimuth tile count and the maximum point count per tile constraint $M$. We use the azimuth tile count to uniformly tile points along the azimuth axis. This procedure happens once per sensor definition and we reuse the same tiling across rasterization calls. We provide pseudo-code in Procedure 1.

**Ray-based Culling.** Compared to cameras, LiDAR scans consist of fewer rays that cover a wider field of view. This results in a sparser pattern where naive tile-based rendering suffers from evaluating numerous Gaussians that that do not contribute to any rays. To address this, we use two separate tile resolutions for rendering and culling $T_r$ and $T_c$, adopting a denser resolution for the latter to cull as many particles as possible ($r_\phi^d = 1600$, $r_\omega^d = 8$). After projecting a particle to the LiDAR angle space, we compute its extent in $T_c$, check if it is intersected by any rays, and only include the particle for rendering (using its coarse extent in $T_r$) if so. We filter in constant time (4 memory reads + 3 arithmetic operations) via a 2D range query where we first construct a 2D ray mask indicating whether rays intersect each culling tile and then build a summed-area table using 2D prefix sum. We provide details in Sec. B of the appendix.

**Beam Divergence.** LiDAR beams have a larger footprint than that of camera rays and diverge significantly as they travel away from the sensor. This causes "bloating" artifacts that are noticeable around reflective surfaces such as traffic signs where only a small portion of the beam hitting the surface is sufficient to obtain a return. These spurious returns (that a true zero-thickness ray would "miss") negatively impact the learned scene geometry when unaccounted for. To address this, we use a 3D smoothing filter similar to AAA-Gaussians (Steiner et al., 2025) but with an important modification to avoid degrading depth. We provide details in Sec. C of the appendix.

### 3.4 OPTIMIZATION

We jointly optimize the camera particles $G_c$, LiDAR particles $G_l$, bilateral grids $\mathcal{A}$, and the environment map by sampling a random input image and LiDAR scan at each training step. We minimize a reconstruction loss, an anchoring loss that encourages camera Gaussians in $G_c$ to lie near the LiDAR-supervised scene geometry distilled into $G_l$, and lower-level regularization terms such that the final loss is $\mathcal{L} := \mathcal{L}_{recon} + \lambda_{anchor}\mathcal{L}_{anchor} + \mathcal{L}_{reg}$, with $\lambda_{anchor} = 0.01$.

**Reconstruction Losses.** We minimize $\mathcal{L}_{recon}$ via L1 photometric loss $\mathcal{L}_{\text{photo}}$, SSIM loss $\mathcal{L}_{\text{SSIM}}$, L1 distance loss $\mathcal{L}_{\text{dist}}$, L1 intensity loss $\mathcal{L}_{\text{int}}$, and binary cross-entropy ray drop loss $\mathcal{L}_{\text{rd}}$:

$$\mathcal{L}_{recon} = \underbrace{\left(\lambda_{photo}\mathcal{L}_{photo} + \lambda_{SSIM}\mathcal{L}_{SSIM}\right)}_{\text{camera losses}} + \underbrace{\left(\lambda_{dist}\mathcal{L}_{dist} + \lambda_{int}\mathcal{L}_{int} + \lambda_{rd}\mathcal{L}_{rd}\right)}_{\text{LiDAR losses}}, \quad (4)$$

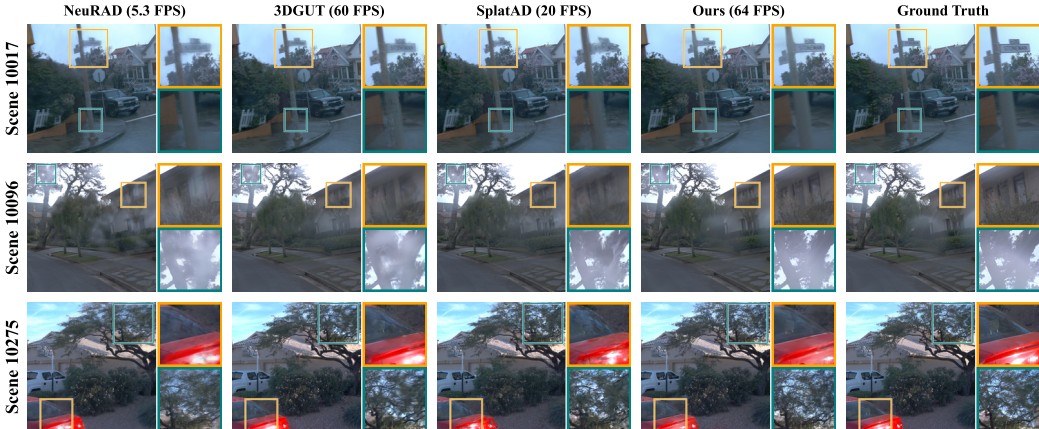

Figure 5: **Static NVS.** Projecting LiDAR as a sparse depth map causes inaccuracies that degrade 3DGUT's rendering of the pole (**above**), which we avoid by rendering LiDAR directly. Our bilateral grids also prevent floaters (**middle**), allowing us to render slightly faster. SplatAD's CNN does not properly recover the sign lettering (**above**) or rear car seat (**below**), while our simpler pipeline does.

Table 1: **Waymo Interp.** SimULi renders the fastest, outperforms all baselines by >2dB PSNR, and gives better depth reconstruction than LiDAR-only LiDAR-RT (Zhou et al., 2025).

| Method | PSNR↑ | SSIM↑ | LPIPS↓ | MedL2.↓ | MeanRelL2↓ | Int. RMSE↓ | RayDrop↑ | CD↓ | MP/s↑ | MR/s↑ |
|---|---|---|---|---|---|---|---|---|---|---|
| Nerfacto (Tancik et al., 2023) | 21.79 | 0.516 | 0.734 | 0.164 | 0.035 | — | — | 0.523 | 0.77 | 0.80 |
| 3DGRT (Moenne-Loccoz et al., 2024) | 27.49 | 0.860 | 0.264 | 0.008 | 0.009 | — | — | 0.196 | 3.23 | 1.39 |
| 3DGUT (Wu et al., 2025b) | 27.23 | 0.856 | 0.271 | 1.252 | 0.206 | — | — | 2.698 | 138.47 | — |
| OmniRe (Chen et al., 2025) | 26.68 | 0.833 | 0.256 | 0.173 | 0.167 | — | — | 0.560 | 24.15 | — |
| StreetGS (Yan et al., 2024) | 26.59 | 0.832 | 0.256 | 0.145 | 0.180 | — | — | 0.566 | 24.85 | — |
| PVG (Chen et al., 2023) | 27.19 | 0.838 | 0.303 | 11.97 | 2.148 | — | — | 3.766 | 6.65 | — |
| DeformableGS (Yang et al., 2024) | 27.95 | 0.856 | 0.235 | 0.090 | 0.169 | — | — | 0.523 | 5.27 | — |
| LiDAR-RT (Zhou et al., 2025) | — | — | — | 0.005 | 0.016 | 0.065 | **0.962** | 0.169 | — | 1.39 |
| UniSim (Yang et al., 2023b) | 23.17 | 0.756 | 0.369 | 0.056 | 0.041 | 0.077 | 0.823 | 0.456 | 8.32 | 0.98 |
| AlignMiF (Tang et al., 2024) | 24.22 | 0.777 | 0.488 | 0.005 | 0.036 | 0.064 | 0.904 | 0.148 | 0.20 | 0.20 |
| NeuRAD (Tonderski et al., 2024) | 27.49 | 0.810 | **0.227** | 0.005 | 0.049 | **0.061** | 0.907 | 0.165 | 13.21 | 1.62 |
| SplatAD (Hess et al., 2025) | 27.82 | 0.839 | 0.246 | 0.008 | 0.013 | **0.061** | 0.916 | 0.175 | 49.98 | 2.40 |
| **SimULi** | **30.15** | **0.881** | 0.241 | **0.003** | **0.007** | 0.064 | 0.944 | **0.136** | 156.90 | 11.33 |

with $\lambda_{photo} = 0.8$, $\lambda_{SSIM} = 0.2$, $\lambda_{dist} = 0.01$, $\lambda_{int} = 0.1$, and $\lambda_{rd} = 0.05$. Camera losses only require rendering and backpropagating through particles in $G_c$ (and LiDAR losses through $G_l$).

**Anchoring Loss.** Existing methods encode camera and LiDAR data into a single NeRF (Yang et al., 2023b; Tonderski et al., 2024) or set of 3D Gaussian particles (Hess et al., 2025). As cross-sensor data contains inconsistencies that are impossible to eliminate, this forces the representation to prioritize the reconstruction quality of one modality over the other based on loss weights (Fig. 4).

By encoding each sensor into its own particle set against which we apply separate reconstruction losses, avoiding this tradeoff is trivial. We still wish to improve camera novel view synthesis via LiDAR-constrained geometry, and thus minimize the distance of camera Gaussians from the predicted surfaces distilled into $G_l$. We define a nearest-neighbor anchoring loss on Gaussian means:

$$\mathcal{L}_{anchor} = \frac{1}{n} \sum_{i \in G_c}^{n} \|\boldsymbol{\mu}_i - NN(\boldsymbol{\mu}_i, G_l)\|_2, \quad NN(\boldsymbol{\mu}, G) = argmin_{\boldsymbol{\mu}' \in G} \|\boldsymbol{\mu} - \boldsymbol{\mu}'\|_2 \quad (5)$$

Since running full nearest-neighbors on all Gaussians in $G_c$ and $G_l$ is computationally expensive, we assign each Gaussian in $G_c$ to its $K = 50$ nearest neighbors in $G_l$. We minimize the K-distance loss at each training iteration, and update the camera-to-LiDAR assignments every 1000 iterations. Compared to the direct LiDAR-supervised depth loss used in prior work, this strategy allows the model a greater degree of freedom in addressing cross-sensor inconsistencies while still improving camera rendering via LiDAR-constrained scene geometry. We ablate its effectiveness in Sec. 4.3.

**Regularization.** As LiDAR is typically reflected by surfaces, we apply an entropy loss to $G_l$ that encourages binary opacity and sparsification. To generate plausible renderings at novel timestamps,

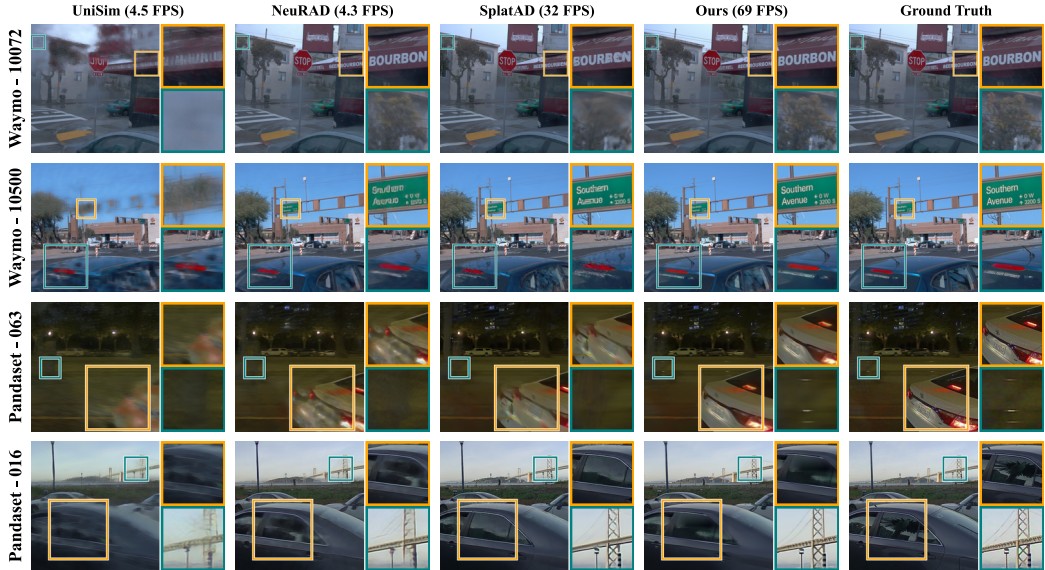

Figure 6: **Dynamic Scenes.** FPS numbers are averaged across *Waymo Dynamic* and PandaSet. Approaches that use CNNs for upsampling (Yang et al., 2023b; Tonderski et al., 2024) or view dependence (Hess et al., 2025) struggle with blurriness and incorrect lettering (**first two rows**). Our rolling shutter strategy handles rapid movement that SplatAD's does not (**third row**). We capture background details and reflections (**bottom**) better than other methods.

Table 2: **Waymo Dynamic.** As with static reconstruction (Table 1), we render the fastest and report best or next-best results across every camera and LiDAR metric.

| Method | PSNR↑ | SSIM↑ | LPIPS↓ | MedL2.↓ | MeanRelL2↓ | Int. RMSE↓ | RayDropAcc↑ | CD↓ | MP/s↑ | MR/s↑ |
|---|---|---|---|---|---|---|---|---|---|---|
| OmniRe (Chen et al., 2025) | 29.45 | 0.897 | 0.162 | 1.188 | 1.013 | — | — | 0.986 | 44.56 | — |
| StreetGS Yan et al. (2024) | 29.60 | 0.898 | 0.161 | 1.280 | 0.739 | — | — | 1.361 | 52.34 | — |
| PVG Chen et al. (2023) | 28.83 | 0.884 | 0.220 | 4.53 | 3.543 | — | — | 3.061 | 15.26 | — |
| DeformableGS Yang et al. (2024) | 28.82 | 0.898 | 0.173 | 1.898 | 0.697 | — | — | 1.299 | 12.90 | — |
| LiDAR-RT Zhou et al. (2025) | — | — | — | _0.003_ | 0.046 | 0.056 | **0.946** | _0.176_ | — | 0.93 |
| UniSim Yang et al. (2023b) | 24.70 | 0.799 | 0.247 | 0.023 | 0.055 | 0.072 | 0.823 | 0.671 | 8.77 | 1.06 |
| NeuRAD Tonderski et al. (2024) | 29.61 | 0.853 | **0.148** | 0.005 | 0.081 | 0.054 | 0.875 | 0.199 | 9.89 | 1.19 |
| SplatAD (Hess et al., 2025) | _30.60_ | _0.900_ | _0.150_ | 0.008 | _0.025_ | _0.055_ | 0.866 | 0.223 | _52.28_ | _2.94_ |
| **SimULi** | **32.35** | **0.922** | _0.150_ | **0.002** | **0.019** | **0.053** | _0.932_ | **0.148** | **179.45** | **10.56** |

we enforce smoothness across our affine color transformations $\mathcal{A}$ and background, and regularize Gaussian scale and opacity as in MCMC (Kheradmand et al., 2024). We provide details in Sec. D.

## 4 EXPERIMENTS

We validate the effectiveness of our method on two commonly used AV datasets across camera-only, LiDAR-only, and joint camera-LiDAR baselines. To measure against the widest possible set of baselines, we first measure novel view synthesis on static scenes used to evaluate prior work (Huang et al., 2023) (Sec. 4.1). As dynamic simulation is our main focus, we evaluate both reconstruction and novel view synthesis in Sec. 4.2. We validate the efficacy of our components in Sec. 4.3.

### 4.1 STATIC ENVIRONMENTS

**Dataset.** We perform experiments on all four scenes of the *Waymo Interp.* benchmark (Huang et al., 2023) and follow the suggested protocol of holding out every 5th frame for validation. We use the front three cameras and render images and LiDAR scans at full resolution.

**Baselines.** We compare SimULi to Nerfacto (Tancik et al., 2023) as a state-of-the-art NeRF approach and adapt 3DGUT (Wu et al., 2025b) to take LiDAR depth as supervision in the form

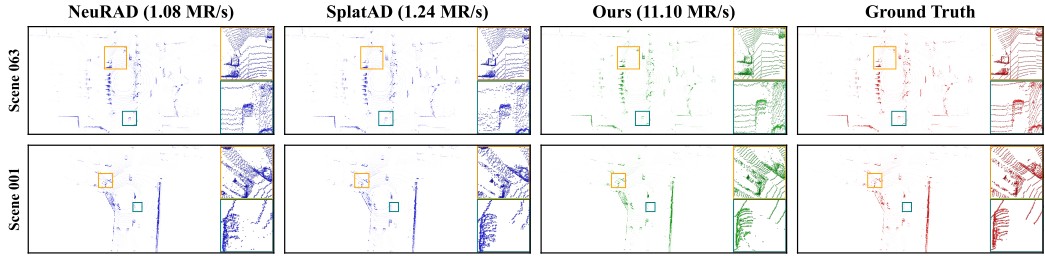

Figure 7: **PandaSet.** SimULi renders the fastest by $>10\times$. Compared to other joint camera-LiDAR methods, ours provides the sharpest LiDAR renderings, especially near vehicles.

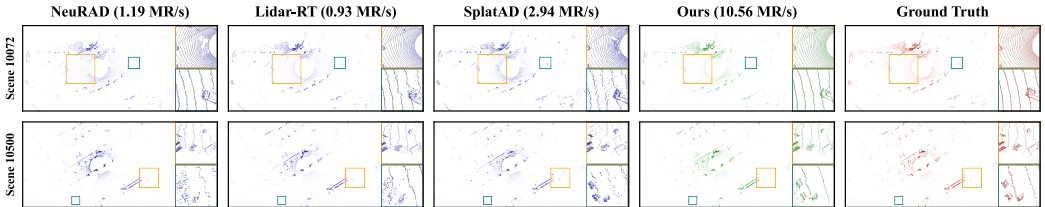

Figure 8: **Waymo Dynamic.** As in Fig. 7, SimULi renders the fastest (as measured on an A40 GPU) and compares favorably to camera-LiDAR and LiDAR-only (Zhou et al., 2025) baselines.

Table 3: **PandaSet Reconstruction.** We outperform all prior work by a wide margin, improving upon the second-best method (SplatAD) by $>1$dB PSNR while rendering camera views 60% faster.

| Method | PSNR↑ | SSIM↑ | LPIPS↓ | MedL2.↓ | MeanRelL2↓ | Int. RMSE↓ | RayDrop↑ | CD↓ | MP/s↑ | MR/s↑ |
|---|---|---|---|---|---|---|---|---|---|---|
| OmniRe (Chen et al., 2025) | 25.82 | 0.779 | 0.322 | 0.131 | 0.083 | — | — | 0.970 | 56.03 | — |
| StreetGS (Yan et al., 2024) | 25.82 | 0.780 | 0.319 | 0.162 | 0.079 | — | — | 1.012 | 68.31 | — |
| PVG (Chen et al., 2023) | 24.71 | 0.723 | 0.464 | 39.859 | 0.593 | — | — | 6.763 | 24.88 | — |
| DeformableGS (Yang et al., 2024) | 25.00 | 0.770 | 0.349 | 0.359 | 0.159 | — | — | 0.289 | 16.52 | — |
| UniSim (Yang et al., 2023b) | 23.53 | 0.693 | 0.334 | 0.047 | 0.079 | 0.087 | 0.917 | 1.326 | 10.05 | 1.07 |
| NeuRAD (Tonderski et al., 2024) | 26.50 | 0.767 | 0.241 | 0.006 | 0.054 | 0.055 | 0.973 | 0.321 | 9.71 | 1.10 |
| SplatAD (Hess et al., 2025) | 28.58 | 0.878 | **0.186** | 0.010 | 0.032 | 0.054 | 0.974 | 0.336 | 86.74 | 1.22 |
| **SimULi** | **29.76** | **0.881** | 0.195 | **0.002** | **0.006** | **0.034** | **0.997** | 0.206 | 137.74 | 11.10 |

of sparse depth maps. We also compare to 3DGRT (Moenne-Loccoz et al., 2024), AV-centric methods in `drivestudio` (Chen et al., 2025; Yan et al., 2024; Chen et al., 2023; Yang et al., 2024) and LiDAR-RT (Zhou et al., 2025) as a LiDAR-only particle ray tracing method. We further measure SplatAD (Hess et al., 2025), NeuRAD (Tonderski et al., 2024) and `neurad-studio`'s UniSim (Yang et al., 2023b) implementation as joint camera-LiDAR baselines, along with the official AlignMiF (Tang et al., 2024) implementation as a NeRF-based method that also addresses camera-LiDAR misalignment in static scenes.

**Metrics.** We evaluate image quality through PSNR, SSIM (Wang et al., 2004), and the AlexNet variant of LPIPS (Zhang et al., 2018). We list the median absolute depth error, mean relative depth accuracy, and chamfer distance of LiDAR predictions in meters, and intensity and ray drop accuracy for methods that support it. As 3DGRT and LiDAR-RT need ray tracing cores, we measure camera and LiDAR rendering speed (denoted in millions of pixels/rays per second) on an NVIDIA A40.

**Results.** We present results in Table 1 and qualitative examples in Fig. 5. Visually, SimULi outperforms all methods by $>2$dB PSNR, and is best or nearly-best across all other metrics. It outperforms LiDAR-RT (Zhou et al., 2025), which solely targets LiDAR reconstruction, on all metrics except ray drop accuracy (for which LiDAR-RT uses a U-Net refinement network) and renders $>10\times$ faster. Although our camera rendering builds upon 3DGUT (Wu et al., 2025b), the improvements in our pipeline (anchoring, appearance variation, environment map) improve rendering quality by 3dB PSNR and produce fewer "floaters", leading to slightly faster rendering. Compared to Align-MiF (Tang et al., 2024), which addresses camera-LiDAR misalignment via feature fusion that incurs a large inference-time cost, our factorization actually improves rendering speed (as we only rasterize the subset of Gaussians relevant to the sensor being rendered).

Table 4: **PandaSet NVS.** Similar to Table 3, SimULi renders the fastest, provides the best PSNR, and outperforms next-best method SplatAD across every LiDAR metric.

| Method | PSNR↑ | SSIM↑ | LPIPS↓ | MedL2.↓ | MeanRelL2↓ | Int. RMSE↓ | RayDrop↑ | CD↓ | MP/s↑ | MR/s↑ |
|---|---|---|---|---|---|---|---|---|---|---|
| OmniRe (Chen et al., 2025) | 25.13 | 0.757 | 0.351 | 0.425 | 0.113 | — | — | 1.126 | 53.19 | — |
| StreetGS (Yan et al., 2024) | 25.09 | 0.756 | 0.352 | 0.378 | 0.102 | — | — | 1.218 | 64.67 | — |
| PVG (Chen et al., 2023) | 24.00 | 0.709 | 0.462 | 30.686 | 0.563 | — | — | 5.793 | 23.68 | — |
| DeformableGS (Yang et al., 2024) | 23.68 | 0.720 | 0.346 | 0.190 | 0.182 | — | — | 1.057 | 15.81 | — |
| UniSim (Yang et al., 2023b) | 23.50 | 0.693 | 0.328 | 0.065 | 0.120 | 0.087 | 0.919 | 1.438 | 11.25 | 0.87 |
| NeuRAD (Tonderski et al., 2024) | 25.97 | 0.759 | 0.243 | 0.009 | 0.075 | 0.062 | 0.963 | 0.360 | 11.37 | 1.08 |
| SplatAD (Hess et al., 2025) | 26.73 | 0.815 | **0.193** | 0.011 | 0.035 | **0.059** | 0.966 | 0.346 | 88.79 | 1.24 |
| **SimULi** | **27.12** | **0.830** | 0.220 | **0.006** | **0.018** | 0.059 | **0.970** | **0.331** | **136.33** | **11.02** |

## 4.2 DYNAMIC ENVIRONMENTS

**Datasets and Baselines.** We measure reconstruction on the same PandaSet scenes as SplatAD (Hess et al., 2025) and novel view synthesis on both PandaSet and the *Waymo Dynamic* benchmark (Wu et al., 2024). We compare to the methods in Sec. 4.1 that handle dynamics.

**Metrics.** We evaluate *Waymo Dynamic* with the same metrics used in Sec. 4.1. On PandaSet, we measure timings on 80GB NVIDIA A100 GPUs to ensure that our comparisons are as close as possible to the setting described in SplatAD (Hess et al., 2025), the baseline closest to our work.

**Results.** We list reconstruction results in Table 3 and novel view synthesis in Tables 2 and 4. We provide camera visuals in Fig. 6 and LiDAR in Figs. 7 and 8. In all cases, SimULi provides the best visual and depth quality. Compared to SplatAD (Hess et al., 2025), the method closest to ours, we improve PSNR by 0.4-1.7 dB without relying on CNNs for view dependence, render cameras 1.5-3× faster, and accelerate LiDAR by 10×.

## 4.3 DIAGNOSTICS

**Representation.** We compare our factorized model to the alternative of encoding all sensors into a single particle set and vary the strength of LiDAR depth loss $\lambda_d$ (Fig. 4). We also ablate the bilateral grids and environment map used to improve camera rendering. We summarize results in Table 5. Not only does anchoring improve NVS compared to camera-only reconstruction ($\lambda_d = 0$), but it outperforms the unified strategy across all metrics for all values of $\lambda_d$, and renders LiDAR 2× faster. Bilateral grids and environment maps improve camera quality to different degrees.

**LiDAR Rendering.** We measure rendering speed on a single scene while varying the maximum number of points per tile $M$ and the number of elevation tiles $N_\phi$. The choice $M = 32, N_\phi = 16$ gives the best LiDAR rendering speed (note that does not affect quality). This highlights the benefits of our automated strategy - we can easily find optimal settings for any sensor via straightforward grid search, instead of manually specifying tile boundaries as in prior work (Hess et al., 2025).

## 5 CONCLUSION

We propose SimULi, the first method that renders arbitrary camera models and LiDAR sensors in real-time. Our automated LiDAR tiling and ray-based culling accelerate rendering >10× relative to prior work. Our factorized representation greatly improves multi-sensor reconstruction quality and is of interest to many downstream applications.

Table 5: **Ablations.** NVS metrics averaged across PandaSet.

| Method | Factorized Gaussians | Bilateral Grid | Env. Map | PSNR↑ | CD↓ | MP/s↑ | MR/s↑ |
|---|---|---|---|---|---|---|---|
| Direct $\lambda_d = 0$ | ✗ | ✓ | ✓ | 26.61 | 6.248 | 134.75 | 5.55 |
| Direct $\lambda_d = 0.001$ | ✗ | ✓ | ✓ | 26.70 | 0.718 | 128.22 | 5.01 |
| Direct $\lambda_d = 0.01$ | ✗ | ✓ | ✓ | 26.39 | 0.475 | 132.45 | 4.97 |
| Direct $\lambda_d = 0.1$ | ✗ | ✓ | ✓ | 25.78 | 0.393 | 131.04 | 4.77 |
| w/o Bilateral Grid | ✓ | ✗ | ✓ | 25.99 | 0.337 | 130.63 | **12.07** |
| w/o Env. Map | ✓ | ✓ | ✗ | 26.81 | 0.336 | 125.30 | 11.83 |
| Full Method | ✓ | ✓ | ✓ | **27.12** | **0.331** | **136.33** | 11.02 |

Table 6: **LiDAR Tiling (MR/s).**

| $N_\phi \setminus M$ | 256 | 128 | 64 | 32 |
|---|---|---|---|---|
| 64 | 6.96 | 7.22 | 12.29 | 14.24 |
| 32 | 5.48 | 7.89 | 13.87 | 13.51 |
| 16 | 6.49 | 12.02 | 13.64 | **15.75** |
| 8 | 9.24 | 9.24 | 10.05 | 15.12 |

## 6 ETHICS STATEMENT

Our approach facilitates the creation and rendering of high-fidelity neural representations. Consequently, the inherent risks mirror those found in other neural rendering research, mainly concerning privacy and security due to the potential capture of sensitive information, whether intentional or not. These concerns are particularly relevant in autonomous vehicle settings, which our method targets, as training data may contain private details like human faces and vehicle license numbers. A potential solution would be to incorporate semantic distillation into the model's representation, as seen in prior work (Kobayashi et al., 2022; Tschernezki et al., 2022; Turki et al., 2023; Kerr et al., 2023), to filter out sensitive data during rendering. However, this information would still persist within the model itself. A more effective mitigation strategy would be to preprocess the input data used to train the model before it is ever ingested (Wang et al., 2017).

## 7 REPRODUCIBILITY STATEMENT

We describe how to build our pipeline in the paper and appendix, provide pseudo-code for our LiDAR tiling and culling strategies, list our training hyperparameters, and detail our experimental protocol. All data used for evaluation is publicly available.

## 8 ACKNOWLEDGMENTS

We thank Nicolas Moenne-Loccoz for assistance with our renderer and the SplatAD authors for discussing their method and evaluation protocol.

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
