# A    LIDAR TILING

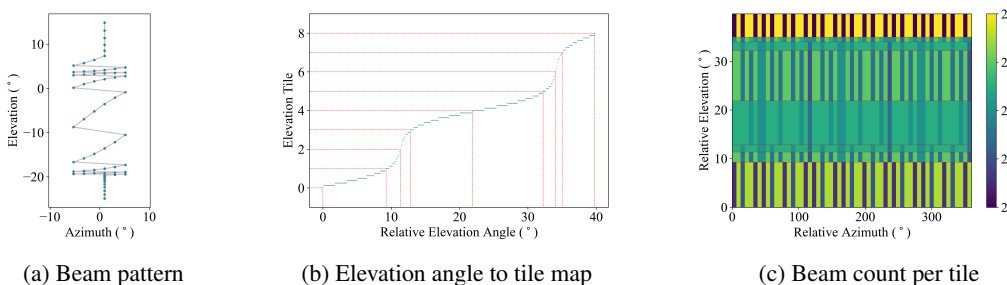

(a) Beam pattern                    (b) Elevation angle to tile map                    (c) Beam count per tile

Figure 9: **Customized LiDAR Model.** We can readily apply our automated tiling strategy to arbitrary spinning LiDAR models.

We describe our LiDAR tiling strategy in Procedure 1. As discussed in Sec. 4.3, a significant advantage of this approach is that it readily be applied to any spinning LiDAR model to find optimal tiling parameters, including the customized model shown in Fig. 9, instead of manual exploration as in prior methods (Hess et al., 2025). While our experiments focus on spinning LiDARs, our method can be extended to non-spinning LiDARs by also applying the tiling procedure along the azimuth dimension according to the sensor's beam pattern.

---

**Algorithm 1** ELEVATIONTILING

---

**Input:** elevation angles: $\mathbf{\Phi}$, elevation tile count $N_\phi$, maximum point count per tile $M$
**Output:** numbers of azimuth tiles: $N_\theta$, elevation tiling: $\boldsymbol{T}$

1:  $r = 400$                                                              ▷ *pre-defined elevation resolution*
2:  $\boldsymbol{H} = \text{Histogram}(\mathbf{\Phi}, \text{nbins} = r)$                           ▷ *compute histogram of elevations*
3:  $\boldsymbol{C} = \text{CumulativeSum}(\boldsymbol{H})$                               ▷ *compute cumulative sum of histogram*
4:  $\boldsymbol{C} = \boldsymbol{C}/\boldsymbol{C}_{\max} \cdot N_\phi$
5:  $\boldsymbol{T} = \{0\}$                                                ▷ *compute elevation tiling iteratively based on $\boldsymbol{C}$*
6:  $b = 1$
7:  **for** $i = 0$ to $N_\phi - 1$ **do**
8:      **if** $\boldsymbol{C}(i) \geq b$ **then**
9:          $\boldsymbol{T}.\text{append}(i)$
10:         $b \mathrel{+}= 1$
11: $\boldsymbol{T} = \boldsymbol{T}/r \cdot (\Phi_{\max} - \Phi_{\min})$                          ▷ *normalize tiling $\boldsymbol{T}$*
12: $\boldsymbol{H} = \text{Histogram}(\mathbf{\Phi}, \text{bins} = \boldsymbol{T})$          ▷ *re-compute histogram based on newly computed tiling $\boldsymbol{T}$*
13: $N_\theta = \boldsymbol{H}_{\max} / M$                     ▷ *compute the number of azimuth tiles based on constraint $M$*
14: **return** $N_\theta, \boldsymbol{T}$

---

Table 7: **Ray-based Culling.** We measure kernel run times in milliseconds. Our filtering incurs a small constant time cost offset by improvements to the subsequent sort and render operations.

| Kernels | w/ Ray Culling | w/o Ray Culling | Speedup (%) |
|---|---|---|---|
| Project | 2.43 | 2.39 | -1.71 |
| Sort | 0.48 | 0.52 | 7.67 |
| Render | 4.71 | 5.17 | 9.02 |

## B  RAY-BASED CULLING

We profile our ray-based culling in Table 7. The constant-time filtering adds a small overhead (about 2%) that is offset by subsequent speedups to sorting and rendering (8-9%). We compute the number of ray occupancies for each particle as described in Procedure 2 and subsequently use it to filter projections as described in Procedure 3.

---

**Algorithm 2** RAYOCCUPANCYCOUNT

**Input:** rectangular extents $E_t$, particle center $p$,
summed-area table of ray mask $S_{\text{ray}}$
**Output:** number of ray occupancies inside the extents $n$
1: $I_{ll} = p - E_t$            ▷*lower left dense tile indices*
2: $I_{ur} = p + E_t$            ▷*upper right dense tile indices*
3:            ▷*Compute ray occupancy counts using the summed-area table*
4: $A = S_{\text{ray}}[I_{ur}.x + 1, I_{ur}.y + 1]$
5: $B = S_{\text{ray}}[I_{ll}.x, I_{ur}.y + 1]$
6: $C = S_{\text{ray}}[I_{ur}.x + 1, I_{ll}.y]$
7: $D = S_{\text{ray}}[I_{ll}.x, I_{ll}.y]$
8: $n = A - B - C + D$
9: **return** $n$

---

**Algorithm 3** PROJECTPARTICLES

**Input:** culling tile extents $E_t$, rendering tile extents $E_r$, particle center $p$,
summed-area table of ray mask $S_{\text{ray}}$
**Output:** coarse tile count $n$
1: $n = 0$
2: **if** RayOccupancyCount($E_t$, $p$, $S_{\text{ray}}$) > 0 **then**
3:      **for** $x = 1$ to $E_r.x$ **do**
4:          **for** $y = 1$ to $E_r.y$ **do**
5:              $n = n + 1$
6: **return** $n$

---

## C   ANTI-ALIASING FOR BEAM DIVERGENCE

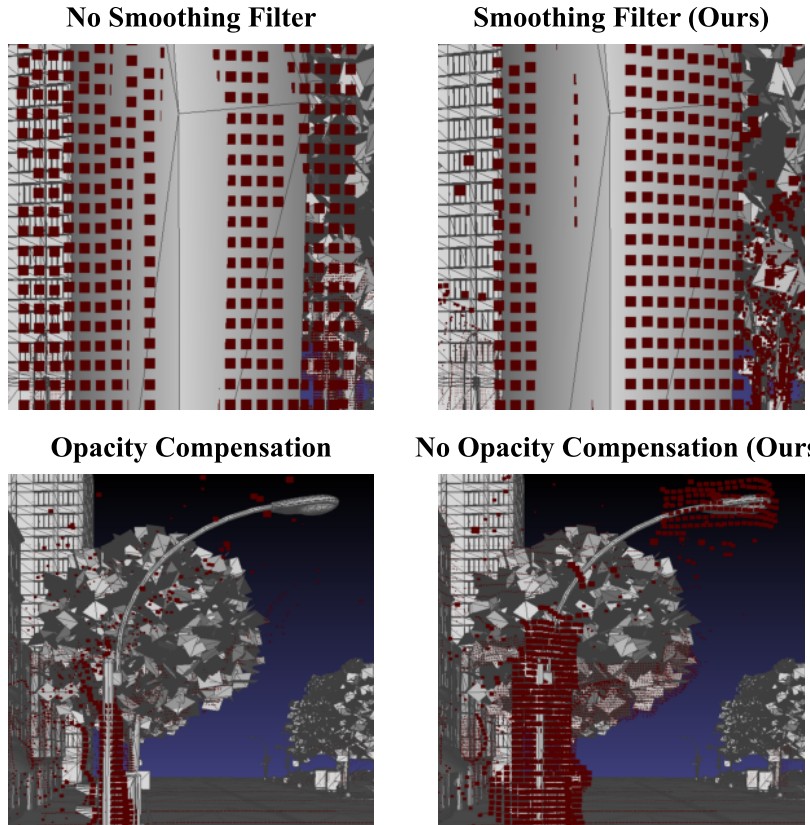

Figure 10: **Beam Divergence.** We show a representative example where LiDAR divergence causes "bloating" (**top-left**) that our filter removes (**top-right**). We crucially omit the opacity compensation used in prior work (Steiner et al., 2025; Yu et al., 2024) as it blends foreground and background objects, adversely affecting the recovered street lamp geometry (**bottom**).

As mentioned in Sec. 3.2, we apply a 3D smoothing filter to LiDAR Gaussians in $G_l$ similar to that used for camera anti-aliasing in AAA-Gaussians (Steiner et al., 2025) that scales its covariance:

$$\hat{\rho}_\perp(\mathbf{x}) = \sqrt{\frac{|\Sigma_\perp|}{|\hat{\Sigma}_\perp|}} \exp(-\frac{1}{2}(\mathbf{x} - \mu)^\top \hat{\Sigma}^{-1}(\mathbf{x} - \mu)) , \tag{6}$$

where $\mathbf{d}$ is the normalized vector between $\mu$ and the camera origin $\mathbf{o}$, and $\Sigma_\perp$ denotes the $2 \times 2$ projected onto the subspace orthogonal to $\mathbf{d}$. However, unlike AAA-Gaussians and prior work (Yu et al., 2024), we do not scale the opacity factor $\sqrt{\frac{|\Sigma_\perp|}{|\hat{\Sigma}_\perp|}}$. Although this change can be seen as straightforward, we found it to have a significant impact on LiDAR reconstruction (and depth data more broadly) as the filter otherwise degrades scene geometry by encouraging blending foreground and background objects. Fig. 10 illustrates the differences in approaches.

## D    IMPLEMENTATION DETAILS

We train our model in the Pytorch framework (Paszke, 2019) and use custom CUDA and Slang (Bangaru et al., 2023) kernels for rendering. We initialize each scene by voxelizing the LiDAR point cloud with a resolution of 0.1 meters. We project the point cloud onto the camera images and initialize the color and scale of each Gaussian with the color and pixel footprint of the closest image. We train on each scene for 30,000 iterations and use a variant of MCMC's densification strategy (Kheradmand et al., 2024) that relocates Gaussians based on reconstruction error instead of opacity. This slightly improves reconstruction quality and combines better with the entropy loss that we apply to Gaussians in $G_l$ (as relocation would otherwise skew towards the binarized high-opacity Gaussians the loss encourages).

To generate plausible renderings at novel timestamps, we encourage smoothness across our affine color transformations $\mathcal{A}$ and environment map via a total variation loss (Wang et al., 2024) $\mathcal{L}_{\text{TV}}$ and identity drift loss $\mathcal{L}_{\text{drift}}$. We also regularize the scale and opacity of our Gaussians as in MCMC (Kheradmand et al., 2024), which we found to improve rendering quality. Our overall regularization is thus:

$$\mathcal{L}_{\text{reg}} := \mathcal{L}_{\text{entropy}} + \mathcal{L}_{\text{TV}} + \lambda_{\text{drift}}\mathcal{L}_{\text{drift}} + \lambda_{\mathbf{\Sigma}} \sum_{ij} \sqrt{\text{eig}_j(\mathbf{\Sigma}_i)} + \lambda_\sigma \sum_i \sigma_i, \tag{7}$$

with $\lambda_{\text{drift}} = 0.001$, $\lambda_{\mathbf{\Sigma}} = 0.001$, and $\lambda_\sigma = 0.005$ in our experiments.

We impose a global cap of 4M Gaussians across both modalities, fewer than the most comparable baseline (SplatAD, 5M). Importantly, our factorized representation improves memory efficiency by (i) reducing the need to introduce "floater" Gaussians to account for cross-sensor inconsistencies and (ii) storing only modality-specific attributes within each Gaussian, whereas a naive unified formulation would redundantly maintain separate camera and LiDAR spherical-harmonic coefficients per primitive. In practice, the resulting model serializes to approximately 900 MB, comparable to prior 3DGS methods, and fits within the memory budgets of contemporary desktop GPUs.

## E    LIMITATIONS

SimULi does not currently model non-rigid objects, although solutions such as Chen et al. (2025) are directly applicable to our method. As with other scene optimization methods, novel view synthesis suffers when rendering viewpoints that greatly deviate from the training poses, which could be mitigated via generative priors as proposed in Wu et al. (2025a); Fischer et al. (2025). Finally, the K nearest neighbors calculations used in the anchoring loss adds a modest training time overhead (around 14 minutes with our default settings). This could be further reduced by adjusting $K$ (setting it to 20 decreases the overhead to 6 minutes with little change in quality) or via fused custom CUDA kernels.

## F    ADDITIONAL VISUALIZATIONS

**Time-dependent Effects.**    As discussed in Sec. 3.2, we approximate each Gaussian by projecting 7 sigma points to estimate its 2D conic. As these points are independently projected, we handle time-dependent effects such as rolling shutter and its LiDAR equivalent at high granularity by incorporating sensor movement into the projection function. We illustrate its impact in Fig. 11.

**LiDAR Noise.**    LiDAR sensors tend to exhibit "bloating" artifacts near thin and reflective surfaces. In prior, non-factorized methods, this noisy "ground-truth" supervision degrades either camera quality (due to learning incorrect LiDAR-supervised geometry) or LiDAR fidelity (since LiDAR loss must be down-weighed and is typically applied uniformly across all points). As illustrated in Fig. 12, factorization provides the flexibility needed to capture these effects without impacting camera quality.

**LiDAR Projection.**    We illustrate the validity of our LiDAR projection strategy by comparing it to Monte Carlo sampling, which is more accurate but expensive to compute, in Fig. 13 and measure its

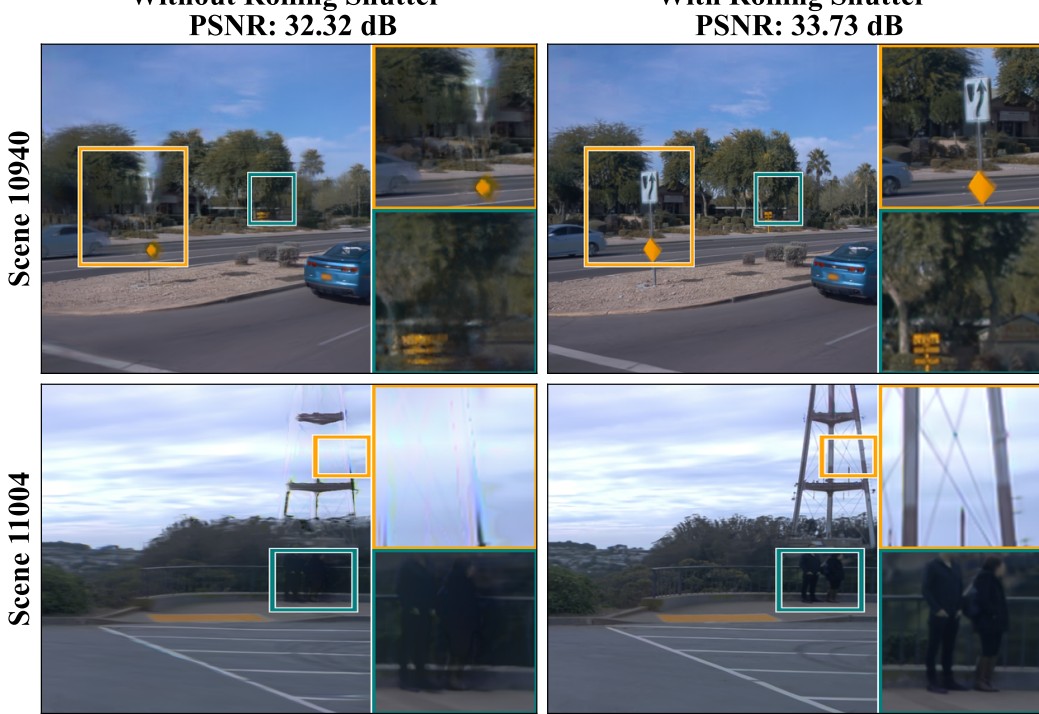

Figure 11: **Rolling Shutter.** Images captured from fast-moving vehicles such as those in the Waymo Open Dataset (Sun et al., 2020) exhibit rolling shutter effects that, when unaccounted for, degrade reconstruction quality (**left**). Our rendering formulation incorporates sensor movement into the projection function and evaluates the particle response function in 3D using the exact timestamp of each ray, allowing us to model time-dependent effects at high granularity (**right**).

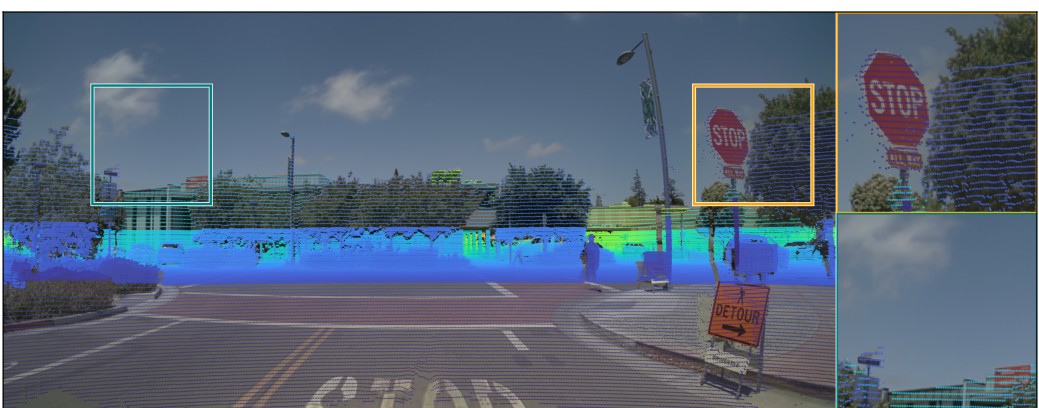

Figure 12: **LiDAR Noise.** LiDAR tends to exhibit "bloating" artifacts near reflective surfaces such as the stop sign (**top inset**) and thin structures (**bottom inset**). Our factorized representation and anchoring loss accurately capture these effects without compromising camera quality.

accuracy in Table 8. Our results are near-identical despite projecting far fewer samples per point (7 vs 200).

Table 8: **LiDAR Projection.** We compare our LiDAR projection strategy to linearization and Monte Carlo sampling alternatives. Our results are near-identical to Monte Carlo sampling despite projecting far fewer points per Gaussian.

| Method | MedL2. ↓ | MeanRelL2↓ | Int. RMSE↓ | RayDrop↑ | CD ↓ |
|---|---|---|---|---|---|
| Linearization | 0.013 | 0.022 | 0.053 | 0.981 | 0.356 |
| Monte Carlo Sampling | **0.003** | **0.011** | **0.037** | **0.991** | 0.252 |
| **SimULi** | **0.003** | **0.011** | **0.037** | **0.991** | **0.250** |

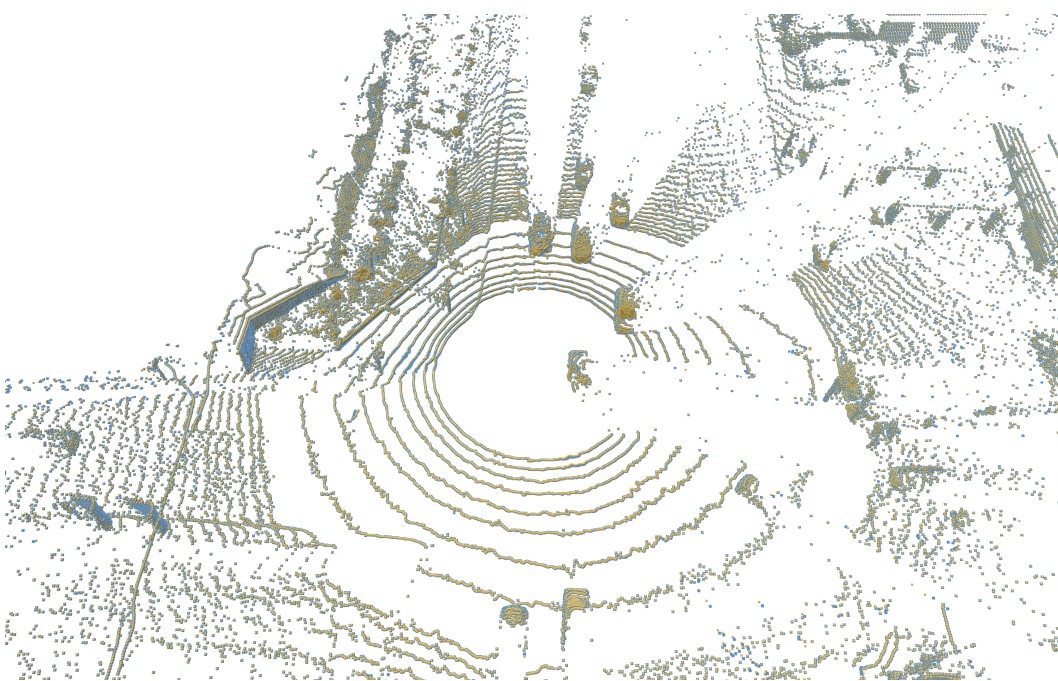

Figure 13: **LiDAR Projection.** We overlay the results of our LiDAR projection function (**orange**) with those generated with Monte Carlo sampling (which is more accurate but expensive to compute) (**blue**). Although our strategy projects far fewer points per Gaussian (7 instead of 200), our results are closely aligned.