# OpenReview forum: "SimULi: Real-Time LiDAR and Camera Simulation with Unscented Transforms"
_ICLR.cc/2026/Conference — ICLR 2026 Poster_

### Official Review · Reviewer_HQsC · 2025-10-29

**Soundness:** 3
**Presentation:** 4
**Contribution:** 3
**Rating:** 8
**Confidence:** 4

**Summary:**

SimULi extends 3DGUT by incorporating LiDAR support, enabling simultaneous modeling of multimodal sensors in autonomous driving scenarios. The system achieves performance on both camera and LiDAR modalities that matches or surpasses the current state of the art.

**Strengths:**

- Achieves joint modeling of multimodal sensors (camera and LiDAR) in autonomous driving scenes, with each modality outperforming existing single-modality methods.

 - Incorporates modeling of fisheye cameras and rolling-shutter effects, demonstrating a highly comprehensive design.

 - Significantly improves rendering speed.

**Weaknesses:**

- The interaction between camera and LiDAR Gaussian primitives relies solely on the anchor loss, which may lead to redundant Gaussian points.

 - The system demonstrates strong engineering merit, but the novelty is relatively limited.

**Questions:**

- How is LiDAR integrated into 3DGUT? Are the seven points projected and weighted following the LiDAR’s ray model? If so, could the authors provide a visualization similar to Fig. 7 in 2DGS to illustrate the validity of this approximation? As nonlinearity of LiDAR projection is typically stronger than that of camera projection.

 - Is the interaction between camera and LiDAR Gaussian primitives only achieved through the anchor loss? Would such a weak coupling lead to redundant Gaussian points?

 - Could the proposed approach be combined with recent generative-aided reconstruction methods (e.g., StreetCrafter, DriveX)?

 - Which component contributes most to the significant improvement in rendering speed?

---

> ### Author Response · Authors · 2025-11-21
>
> Thank you for reviewing our paper! We are glad that you appreciate the strong performance of our method and comprehensiveness of our design, and hope to address any remaining concerns you may have.
>
> **LiDAR projection.** Your understanding is correct - we implement an exact, rolling-shutter-aware, inverse projection model for LiDAR and project 7 sigma points per Gaussian to estimate its 2D conic. To address your concerns about its validity, we visualize it relative to Monte Carlo sampling (which is theoretically more accurate but more expensive to compute) in Figure 12 and Table 8 of the appendix in the latest uploaded version of our paper (Table 8 pasted below for convenience). Our results are near-identical to the more expensive Monte Carlo sampling despite projecting far viewer points per Gaussian (7 vs 200), and significantly more accurate than the linearization strategy used in the original 3DGS.
>
> |Method              |Med. L2 Depth↓|Rel. L2 Depth↓|Intensity RMSE↓|Raydrop Acc.↑|Chamfer Dist.↓  |
> |--------------------|-----|-----|-----|-------|-----|
> |Linearization       |0.013|0.022|0.053|0.981  |0.356|
> |Monte Carlo Sampling|0.003|0.011|0.037|0.991  |0.252|
> |SimULi              |0.003|0.011|0.037|0.991  |0.250 |
>
> **Redundancy.** Although our factorized representation might, at first glance, be perceived as redundant, in practice we observe superior results to unified representations with slightly fewer Gaussians! Concretely, we set the global maximum number of Gaussians in our experiments to 4 million (across both modalities) vs 5 million for the baseline closest to ours (SplatAD). We hypothesize that one reason for this is that we avoid allocating gaussians as "floaters" to explain cross-sensor inconsistencies.
>
> **Rendering speed.** Since we largely avoid neural networks in our rendering pipeline (which differs from SplatAD’s use of MLP/CNN decoders), the most important contributor to rendering speed is the number of Gaussians. Here, our factorized representation is key, since we only need to render LiDAR Gaussians when rendering LiDAR, and camera Gaussians when rendering camera views. Since LiDAR is typically sparse, we end up allocating only about one million LiDAR gaussians per scene (our MCMC relocation strategy allows for Gaussians to flow between modalities, so the exact number varies per scene), which contributes to our large speedups relative to existing SOTA (SplatAD). As shown in Table 6 in our paper, the choice of tiling also has a large impact and is easily tunable with our automated tiling strategy, while the ray-based culling has a more modest effect.
>
> **Generative methods.** Yes, our method should complement approaches such as StreetCrafter! We suspect that our factorized representation would benefit methods that distill diffusion-generated views back into the 3D representation, as it will be more robust to the slight inconsistencies inherent to these models.

---

> > ### Comment · Reviewer_HQsC · 2025-11-24
> >
> > Thank you for the authors’ response. I am happy to maintain my original score. In addition, I believe it would be beneficial to include a visualization similar to Fig. 7 in 2DGS in the final copy, as it provides a direct and intuitive depiction of the Gaussian primitives.

---

> > > ### Author Response · Authors · 2025-11-27
> > >
> > > Dear Reviewer,
> > >
> > > Thank you for your reply - again we really appreciate your consideration and feedback!
> > >
> > > Best,
> > > Authors

---

### Official Review · Reviewer_DQih · 2025-11-01

**Soundness:** 2
**Presentation:** 3
**Contribution:** 3
**Rating:** 6
**Confidence:** 3

**Summary:**

SimULi describes a framework for real-time simulation of LiDAR and camera for autonomous driving scenes. The focus of this work is to improve the rendering speed and support arbitrary LiDAR and camera configurations. To do so, this work builds on 3DGUT to support arbitrary spinning LiDAR configuration. Furthermore, a factorized 3D Gaussian representation and anchoring strategy was proposed to address discrepancies between simulated LiDAR and camera data. The proposed method was benchmarked on two datasets to showcase the fidelity and efficiency in relation to existing works.

**Strengths:**

1.	The paper is well-written and easy to follow. The limitations of prior works and their relation to the proposed work are clearly highlighted.
2.	The proposed method demonstrates strong rendering fidelity and significant boosts in speed compared to state-of-the-art methods.

**Weaknesses:**

1.	The proposed method has been evaluated on a relatively limited set of datasets. Common benchmarks used in prior works, such as nuScenes and Argoverse 2, would help demonstrate the robustness of the method across datasets and sensor setups.
2.	SplatAD encodes all sensor information into the same Gaussian set. SimULi proposes to encode each sensor into its own particle set to address the inconsistencies between LiDAR and camera. The impact of this on the training time and potentially memory requirements is not discussed.

**Questions:**

1.	How does factorizing the Gaussian set impact training speed? How does the training time of the proposed work compare against existing works?
2.	For the anchoring loss, the choice of 50 nearest neighbors and updating assignments every 1000 iterations are hyperparameters. How does varying the number of nearest neighbors and update frequency impact convergence?

---

> ### Author Response · Authors · 2025-11-21
>
> Thank you for reviewing our paper and for the helpful feedback!
>
> **Memory usage.** We set a global maximum of 4 million Gaussians across both modalities, which is less than the baseline closest to ours (SplatAD, which uses 5 million). Our factorized representation actually improves our ability to represent the scene efficiently (instead of being a hindrance), as (1) we avoid assigning Gaussians as "floaters" to explain cross-sensor inconsistencies and (2) each Gaussian only contains the information needed to represent its sensor type (a naive unified representation would store two sets of spherical harmonics for camera and LiDAR within each Gaussian). Our Gaussians serialize to around 900 MB (similar to other 3DGS methods) and easily fits into the memory of contemporary desktop GPUs.
>
> **Training speed.** The main training time difference between our method and contemporary 3DGS methods such as SplatAD come from: (1) applying the anchor loss at each iteration (which measures the distance of each camera Gaussians to its K nearest LiDAR neighbors) and (2) assigning camera gaussians to LiDAR neighbors every N iterations (which triggers a full K-NN calculation between the camera and LiDAR sets). With K=50 and N=1000, and without using custom kernels, the anchor loss takes around 13 ms per iteration (6.5 minutes over 30k iterations), and the full K-NN computation takes 15 seconds (7.5 minutes over 30k iterations), giving an overall training overhead of 14 minutes. This can be further reduced to 6ms per iteration and 6 seconds per assignment (6 minutes total) by setting K=20 with minimal loss in quality.
>
> **Anchoring loss hyperparameters.** Varying K between 20 and 100, and N between 500 and 2000 on our Pandaset NVS benchmark shows little difference (<0.05 db in PSNR). Our default hyperparameters of K=50 and N=1000 are conservative since the training overhead is relatively modest.

---

> > ### Author Response · Authors · 2025-11-27
> >
> > Dear Reviewer,
> >
> > With the discussion period ending in less than a week, we wanted to gently ping and see if our reply addresses your concerns. Thank you again for taking the time to review our paper!
> >
> > Best,
> > Authors

---

### Official Review · Reviewer_u66L · 2025-11-01

**Soundness:** 3
**Presentation:** 3
**Contribution:** 3
**Rating:** 6
**Confidence:** 3

**Summary:**

this paper introduces a simulator method for autonomous driving that can render both complex camera models and LiDAR data in real time. It builds on 3DGUT and improves cross-sensor consistency, making it more accurate and faster than existing methods. The paper conducted experiments on two public datasets and show state-of-the-art performances.

**Strengths:**

1. The paper is well written and easy to follow.
2. The proposed method consistently outperforms baselines according to both visualizations and tables.
3. The proposed method achieves significantly faster rendering speed than baselines.
4. The  factorized representation is interesting and innovative, which improves both the camera and lidar rendering accuracy.

**Weaknesses:**

would the factorized representation largely increase the total number of Gaussians in the scene? Does this lead to significantly higher memory usage compared to unified representations?

**Questions:**

1. From what I understand, the authors equalize the elevation tiling using a 1D CDF of elevation angles, and then reuse the same azimuth tiling across the whole scan. Wouldn’t that implicitly assume that the LiDAR point distribution is separable between elevation and azimuth, and also static over time? In practice, the point density may vary a lot with azimuth (depending on the scene or motion), so a fixed azimuth tiling might lead to load imbalance — some tiles being overloaded and others almost empty. Did you observe this issue in your experiments, or did you use any mechanism to adapt the azimuth tiling dynamically?
2. Since each camera Gaussian is softly constrained to stay close to its nearest LiDAR neighbor, how sensitive is the method to that K-nearest-neighbor choice (K = 50)? Also, do you ever observe issues where the anchoring loss pulls camera Gaussians toward noisy or missing LiDAR points, especially around thin structures or reflective surfaces?

---

> ### Author Response · Authors · 2025-11-21
>
> Thank you for reviewing our submission and for the insightful questions!
>
> **Memory usage.** We use fewer Gaussians in our experiments (capped to 4 million across both modalities) than the baseline closest to ours (SplatAD, which uses 5 million). Our factorized representation improves our ability to represent the scene efficiently instead of being a hindrance, as (1) we avoid assigning Gaussians as "floaters" to explain cross-sensor inconsistencies and (2) each Gaussian only stores in the information needed to represent its modality (a naive unified representation would store two sets of spherical harmonics for camera and LiDAR within each Gaussian). Our Gaussian representation serializes to about 900 MB (similar to other 3DGS methods) and easily fits into the memory of contemporary desktop GPUs.
>
> **Azimuth tiling and point distributions.** Our tiling strategy computes a static 1D CDF based on the intrinsic beam distribution of the LiDAR sensor. We assume that the beam layout remains fixed over time for a given sensor (and precompute the tiling once at the start of training), such that each tile processes approximately the same number of LiDAR rays. Regular azimuth tiling works well with the spinning LiDAR sensors used in our evaluation datasets, but our tiling procedure could be readily adapted to support irregular azimuth tiling via two 1D CDFs if needed (ie: for non-spinning LiDAR).
>
> However, as you correctly note, although we assume that the distribution of LiDAR rays remains static over time, the distribution of Gaussians that intersect these rays as we move in the scene does not! We handle this by constructing per-tile Gaussian lists and early ray termination (as in the original 3DGS), but find that load imbalance is more pronounced for LiDAR rendering. This motivates our ray-based culling strategy and use of finer-resolution LiDAR tiles as described in Sections 3.3 and C of the appendix.
>
> **Anchoring loss.** To explore this, we varied K between 20 and 100 on our PandaSet NVS benchmark and found little difference (<0.05 db in PSNR), suggesting that our method is robust to the specific choice of hyperparameter (we pick K=50 in our experiments as a conservative guess as the training overhead of around 14 minutes is relatively modest).
>
> To answer your question about LiDAR noise near thin and reflective surfaces, we provide an illustrative example in Figure 11 of the appendix in the latest uploaded draft of our paper. In prior, non-factorized methods, when unaccounted for, this noisy ``ground-truth" supervision degrades either camera quality (due to the model learning incorrect LiDAR-supervised geometry) or LiDAR fidelity (since LiDAR loss must be down-weighed and is typically applied uniformly across all points). In comparison, our factorization and anchoring loss grants the flexibility needed to capture these effects without degrading camera quality.

---

> > ### Author Response · Authors · 2025-11-27
> >
> > Dear Reviewer,
> >
> > With the discussion period ending in less than a week, we wanted to gently ping and see if our reply addresses your concerns. Thank you again for taking the time to review our paper!
> >
> > Best,
> > Authors

---

### Official Review · Reviewer_6a2S · 2025-11-05

**Soundness:** 3
**Presentation:** 3
**Contribution:** 2
**Rating:** 4
**Confidence:** 5

**Summary:**

This paper introduces a 3DGS-based simulator for real-time camera and LiDAR rendering in autonomous driving. The main contributions include a non-equidistant tiling strategy that efficiently handles arbitrary spinning LiDAR sensors, and a factorized 3D Gaussian representation that mitigates cross-sensor inconsistencies between camera and LiDAR modalities, thereby improving rendering realism. The experiments are comprehensive, with extensive comparisons on the Waymo and PandaSet datasets, demonstrating state-of-the-art performance and realism.

**Strengths:**

* The paper is well written and clearly motivated.
* Experiments are thorough and cover multiple datasets and baselines. The proposed method achieves strong quantitative and qualitative performance, demonstrating competitive or superior rendering realism and efficiency.

**Weaknesses:**

* The novelty is limited. The method feels like a natural extension of 3DGUT, and the way LiDAR rendering is supported is conceptually similar to SplatAD.
* The improvement in handling the camera–LiDAR accuracy tradeoff mainly comes from the decoupled representation, but the deeper issue—imperfect sensor modeling (e.g., motion blur, rolling shutter, or calibration)—is not really addressed. Prior work such as NeuRAD has shown that explicitly modeling these effects can significantly improve reconstruction quality.
* This paper also omits related efforts such as AlignMiF, which also tackle multimodal alignment in autonomous driving simulation. A discussion or comparison with AlignMiF would make the contribution clearer and better positioned.

**Questions:**

* The paper claims improved cross-sensor consistency, but how does the method perform when modeling more accurate physical effects such as rolling shutter, motion blur, or calibration errors? This could also be evaluated under controlled conditions, for example using CARLA.
* Can the proposed LiDAR tiling strategy generalize to non-spinning LiDAR sensors, such as solid-state LiDARs, where the sampling pattern is different?

---

> ### Author Response · Authors · 2025-11-21
>
> Thank you for reviewing our paper and for your thoughtful feedback.
>
> **AlignMiF.** We agree that it is relevant to our work! We have accordingly noted it as related work in the latest revision of our paper and measured its performance in our Waymo Interp experiments in Table 1 (we omit it from other experiments as it does not support dynamics). Similar to our work, AlignMiF notes the need to resolve cross-sensor inconsistencies when doing joint camera-LiDAR reconstruction. However, it uses a NeRF/iNGP-based architecture and addresses the issue by fusing features across different hashtables (instead of our more explicit K-nearest neighbor-based anchoring loss). Similar to the findings in their paper, AlignMiF provides better visual quality than UniSim (albeit worse than NeuRAD, another NeRF-based method concurrent to theirs) and competitive LiDAR quality (the second-best chamfer distance after ours). The comparison highlights a key advantage of our novel factorization - contrary to feature fusion which carries an inference time overhead (AlignMiF renders the slowest of all methods in our benchmark), our approach improves rendering speed as we only rasterize the subset of Gaussians being rendered. This is especially beneficial for LiDAR which tends to be naturally sparse, as we demonstrate in Section 4.3 and Table 4 of our paper. We list the comparison between our method, theirs, and other NeRF baselines in the table below for convenience.
>
>
> | |PSNR↑ |SSIM↑ |LPIPS↓|Med. L2 Depth↓|Rel. L2 Depth↓|Intensity RMSE↓|Raydrop Acc.↑|Chamfer Dist.↓|Camera MP/s↑|LiDAR MR/s ↑|
> |--------|-----|-----|-----|---------------|-----------------|--------------|-----------------|----------------|----------|----------|
> |UniSim  |23.17|0.756|0.369|0.056          |0.041            |0.077         |0.823            |0.456           |8.32      |0.98      |
> |NeuRAD  |27.49|0.810|0.227|0.005          |0.049            |0.061         |0.907            |0.165           |13.21     |1.62      |
> |AlignMiF|24.22|0.777|0.488|0.005          |0.036            |0.064         |0.904            |0.148           |0.20      |0.20      |
> |SimULi  |30.15|0.881|0.241|0.003          |0.007            |0.064         |0.944            |0.136           |156.90    |11.33     |
>
>
>
> **Sensor modeling.** Our method implements an exact, rolling-shutter-aware, inverse projection model for LiDAR sensors and approximates each Gaussian via the Unscented Transform (as in 3DGUT) as discussed in Sections 3.2 and 3.3. Concretely, we project 7 sigma points per Gaussian to estimate its 2D conic. As these points are independently projected, we handle time-dependent effects such as rolling-shutter (for both camera and LiDAR) at high granularity by incorporating sensor movement into the projection function. This is a more flexible and accurate approach than SplatAD’s, which limits itself to estimating per-Gaussian velocities in 2D pixel space to adjust their projected 2D means (ignoring distortion in other properties such as the projected covariance). Moreover, as SplatAD uses 3DGS’s EWA splatting formulation, these projection errors propagate to the rendering stage, whereas we evaluate particle response functions in 3D world space as in [1, 2], using the exact timestamp of each ray. We provide an illustration in our supplementary materials (rs-demo.mp4), and add another illustration in Figure 10 of the appendix in the latest draft of our paper.
>
> **LiDAR tiling.** Our tiling strategy can somewhat be viewed as a generalized version of SplatAD’s method. Instead of manually deriving tile boundaries for a specific sensor layout, we automatically compute an approximately optimal, load-balanced tiling from arbitrary LiDAR beam patterns as described in Algorithm 1 of the appendix. As illustrated in Section 4.3 and Table 5, we can easily find optimal settings via grid search instead of relying on per-sensor manual tuning, which is helpful in practice given the diversity of possible sensor designs.
>
> **Non-spinning LiDAR.** Our method can be readily adapted to non-spinning LiDARs by also applying the tiling procedure in the azimuth direction according to the given beam pattern.
>
> References:
>
> [1] Nicolas Moenne-Loccoz, Ashkan Mirzaei, et al. 3D Gaussian Ray Tracing: Fast Tracing of Particle Scenes. In SIGGRAPH ASIA 2024.
>
> [2] Qi Wu, Janick Martinez Esturo, et al. 3DGUT: Enabling Distorted Cameras and Secondary Rays in Gaussian Splatting. In CVPR 2025.

---

> > ### Author Response · Authors · 2025-11-27
> >
> > Dear Reviewer,
> >
> > With the discussion period ending in less than a week, we wanted to gently ping and see if our reply addresses your concerns. Thank you again for taking the time to review our paper!
> >
> > Best,
> > Authors

---

### Author Response · Authors · 2025-11-21

We appreciate our reviewers' thoughtful feedback and are glad they universally concur that our paper is "well-written and easy to follow" (6a2S, DQih). They agree that our method demonstrates "strong quantitative and qualitative performance" (6a2S) and "matches or surpasses the current state of the art" (HQsC), that our "experiments are thorough and cover multiple datasets and baselines" (6a2S), and that our "factorized representation is interesting and innovative" (u66L).

A common concern shared by reviewers (u66L, DQih, HQsC) is whether our factorized representation increases the total number of Gaussians in the scene (affecting memory and training time). In our experiments we cap the overall number of Gaussians across both sensor types to 4 million, which is slightly less than the baseline closest to ours (SplatAD, which uses a maximum of 5 million). We hypothesize that factorizing actually improves our ability to encode the scene efficiently instead of being a hindrance, as (1) we avoid allocating Gaussians as "floaters" to explain cross-sensor inconsistencies and (2) each Gaussian only stores in the information needed to represent its modality (a naive unified representation would store two separate sets of spherical harmonics for camera and LiDAR within each Gaussian). Concretely, our Gaussians serialize to about 900 MB (inline with other 3DGS methods) and could be further optimized via complementary 3DGS compression methods such as vector quantization [1, 2] and visibility filtering [3].

Another common question was how the choice of K used in the K-nearest neighbor anchoring loss (u66L, DQiH) and the frequency of neighbor assignments (DQiH) affects convergence and training speed. Although we set K=50 as a conservative best guess in our experiments (and update the assignments every N=1000 iterations), varying K between 20 and 100 and N between 500 and 2000 results in negligible differences (<0.05 db in PSNR). As to training time, applying the anchoring loss with K=50 takes 13ms per iteration (6.5 minutes over 30k iterations) and the full K-NN assignment takes 15 seconds every N=1000 iterations (7.5 minutes total). Choosing K=20 reduces the anchoring loss overhead to 6ms per iteration and 6 seconds per assignment (6 minutes total). The cost is relatively minor in both cases, and our training speed is otherwise comparable to other 3DGS-based pipelines. We have updated the latest version of our submission to note these considerations, along with other feedback that we address in reviewer-specific comments.

References:

[1] K L Navaneet et al. CompGS: Smaller and Faster Gaussian Splatting with Vector Quantization. In ECCV 2024.

[2] Zhiwen Fan, Kevin Wang, et al. LightGaussian: Unbounded 3D Gaussian Compression with 15x Reduction and 200+ FPS. In NeurIPS 2024.

[3] Michael Niemeyer et al. RadSplat: Radiance Field-Informed Gaussian Splatting for Robust Real-Time Rendering with 900+ FPS. In 3DV 2025.

---

### Meta-Review · Area_Chair_Nx3g · 2025-12-30

**Summary:**

The reviewers’ concerns mainly focus on the novelty of the method, including comparisons with works such as 3DGUT, NeuRAD, and AlignMiF, as well as issues related to efficiency and memory usage.

**Reviewer Concerns:**

Most of the concerns have been addressed in the rebuttal. Some additional visualizations and quantitative comparisons are expected to be included in the final version. The authors are encouraged to include additional experiments and explanations comparing their method with closely related works, as discussed in the rebuttal.

**Reviewer Scores:**

The reviewers 6a2S may raise his scores after the discussion, since most of his concerns are addressed. The other reviewers tend to maintain their original scores.

---

### Decision · Program_Chairs · 2026-01-26

Accept (Poster)